# Functional Diversity of the Litter-Associated Fungi from an Oxalate-Carbonate Pathway Ecosystem in Madagascar

**DOI:** 10.3390/microorganisms9050985

**Published:** 2021-05-01

**Authors:** Vincent Hervé, Anaële Simon, Finaritra Randevoson, Guillaume Cailleau, Gabrielle Rajoelison, Herintsitohaina Razakamanarivo, Saskia Bindschedler, Eric Verrecchia, Pilar Junier

**Affiliations:** 1Laboratory of Microbiology, Institute of Biology, University of Neuchâtel, 2000 Neuchâtel, Switzerland; anaele.simon@gmail.com (A.S.); guillaume.cailleau@unine.ch (G.C.); saskia.bindschedler@unine.ch (S.B.); pilar.junier@unine.ch (P.J.); 2Laboratory of Biogeosciences, Institute of Earth Surface Dynamics, University of Lausanne, 1015 Lausanne, Switzerland; finaritraran@gmail.com (F.R.); eric.verrecchia@unil.ch (E.V.); 3Max Planck Institute for Terrestrial Microbiology, 35043 Marburg, Germany; 4Ecole Supérieure des Sciences Agronomiques, Université d’Antananarivo, Antananarivo 101, Madagascar; g.rajoelison@yahoo.fr; 5Laboratoire des Radio-Isotopes, Université d’Antananarivo, Antananarivo 101, Madagascar; herintsitohaina.razakamanarivo@gmail.com

**Keywords:** *Tamarindus indica*, calcium oxalate crystals, oxalotrophy, litter, carbon cycle, oxalogenic tree, Kirindy forest

## Abstract

The oxalate-carbonate pathway (OCP) is a biogeochemical process linking oxalate oxidation and carbonate precipitation. Currently, this pathway is described as a tripartite association involving oxalogenic plants, oxalogenic fungi, and oxalotrophic bacteria. While the OCP has recently received increasing interest given its potential for capturing carbon in soils, there are still many unknowns, especially regarding the taxonomic and functional diversity of the fungi involved in this pathway. To fill this gap, we described an active OCP site in Madagascar, under the influence of the oxalogenic tree *Tamarindus indica*, and isolated, identified, and characterized 50 fungal strains from the leaf litter. The fungal diversity encompassed three phyla, namely Mucoromycota, Ascomycota, and Basidiomycota, and 23 genera. Using various media, we further investigated their functional potential. Most of the fungal strains produced siderophores and presented proteolytic activities. The majority were also able to decompose cellulose and xylan, but only a few were able to solubilize inorganic phosphate. Regarding oxalate metabolism, several strains were able to produce calcium oxalate crystals while others decomposed calcium oxalate. These results challenge the current view of the OCP by indicating that fungi are both oxalate producers and degraders. Moreover, they strengthen the importance of the role of fungi in C, N, Ca, and Fe cycles.

## 1. Introduction

Microorganisms are essential drivers of plant material degradation and recycling in terrestrial ecosystems [1], and as such, they play key roles in numerous biogeochemical cycles. Soil fungi and bacteria are particularly well known to be involved in plant carbohydrate degradation [2,3]. While most studies focused on carbon, nitrogen, and phosphorus cycles [4,5], other elements, such as calcium, have received much less attention [6,7]. The coupling between carbon and calcium cycles is well described in the hydro-atmosphere system [8]. However, these two elements are also intimately linked in terrestrial ecosystems, including forest soils [9,10,11], and microorganisms contribute to various processes and steps of these cycles [12,13,14,15]. This is, for instance, the case in the oxalate-carbonate pathway (OCP).

The OCP is a biogeochemical process involving plant, fungal, and bacterial partners [16]. In this pathway, various plants producing oxalic acid eventually leads to the formation of calcium oxalate (Caox) crystals when the pH exceeds pH 4.5. Above this value, oxalic acid (p*K_a_*_1_ = 1.27; p*K_a_*_2_ = 4.27) forms its conjugate base, namely oxalate, which can bind to calcium cations. The resulting Caox crystals can be found in leaf, wood, and root tissues, as well as in the surrounding soil [17,18]. Similarly, saprotrophic fungi involved in plant biomass degradation also contribute to the production and release of Caox crystals [19,20]. These crystals can then be degraded by oxalotrophic bacteria [21], leading to a local increase in pH. If the pH reaches pH 8.4, corresponding to the stability of calcium carbonate (CaCO_3_) under environmental conditions, this can lead to the formation of a CaCO_3_ phase. The OCP has been observed mainly in tropical ecosystems, and this process has been reported to occur in soil [22] and in the bark of trees [23].

The OCP has recently received a lot of interest for its potential role in the capture and fixation of carbon in a mineral form in soils [24,25,26], but many questions remain open. In this system, the current paradigm for the functional role of each group of organisms is that both plants and fungi are oxalate-producers (i.e., oxalogenic) and that bacteria are oxalate-consumers (i.e., oxalotrophic) [13,20]. Presently, the taxonomy of oxalotrophic bacteria [21,23,27,28,29] and oxalogenic plants [23,30] has been investigated and documented. However, to date, information about the diversity of oxalogenic fungi is scarce [31,32]. Indeed, there is no data about the taxonomic and functional diversity of the fungi associated with the OCP. Besides their role as oxalate-producers, little is known about the functions of these fungi in the OCP. In order to fill this gap, we used a culture-dependent approach to isolate, identify, and characterize fungal strains from the litter of an OCP ecosystem in Madagascar. These strains were grown on various artificial media to study their functional potential. Our study aims at: (*i*) isolating and phylogenetically identifying fungi associated with an active OCP, (*ii*) characterizing the functional potential of these fungal strains, especially regarding their contributions to C, N, P, Ca and Fe cycles, and (*iii*) clarifying the role of fungi in the functioning of the OCP.

## 2. Materials and Methods

### 2.1. Sampling

We sampled litter of *Tamarindus indica* in November 2014, in the Kirindy forest, Madagascar (coordinates 20°04.515′ S 44°40.236′ E, altitude 68 m). Mean annual temperature and mean annual precipitation were 24.8 °C and 785 mm, respectively. We collected the litter in 50 mL sterile Falcon tubes, using sterile tweezers to prevent biological contamination. We only sampled the upper litter in order to avoid contamination from soil organisms. The sampled litter was stored at 4 °C until laboratory analysis. Additionally, we dug and sampled four soil profiles, three at the base of the *T. indica* trunk and another at 15 m away from the tree. The latter was considered as not under the influence of other oxalogenic plants. Each profile was 80 cm deep. For each profile, we sampled 50 g of soil from 5 cm layers, at four depths: 0–5 cm, 10–15 cm, 20–25 cm, and 40–45 cm.

### 2.2. Soil and Litter Analyses

We measured soil pH H_2_O with a Metrohm^TM^ pH-meter, using 10 g of 2 mm sieved soil mixed with 25 mL of deionized water and agitated for 16 h [22]. Litter was crushed and powdered (5–10 µm) using a Pulverisette 9 (Fritsch, Welden, Germany). Then we determined its mineralogical content using an ARL Xtra diffractometer (Thermo, Waltham, MA, USA). We analyzed the diffractograms using the MacDiff software and converted peak intensities of major minerals into relative abundances [33].

### 2.3. Isolation of the Fungal Strains

To cover a broader range of microenvironments, we used two approaches for fungal isolation. First, we sampled various litter fragments under sterile conditions and washed them 3 times with sterile water by vortexing for 5 min. Subsequently, we directly inoculated the litter fragments in Petri dishes containing lignocellulose agar (LCA) medium with 2 antibiotics (chloramphenicol 0.01% and streptomycin 0.01%). LCA medium contains glucose 0.1%; KH_2_PO_4_ 0.1%; MgSO_4_∙7H_2_O 0.02%; KCl 0.02%; NaNO_3_ 0.2%; yeast extract 0.02% and agar 1.3% (*w*/*v*) [34]. The pH of the medium was adjusted to 5. It is noteworthy that LCA does not contain lignin or cellulose or hemicelluloses [34]. Second, we prepared microbial suspensions in which 15 g of litter were mixed with 150 mL of sterile water, and then, we made a slurry using a sterilized Waring blender. Subsequently, we prepared six serial dilutions, ranging from 10^0^ to 10^−5^. We plated fifty µL of each dilution in Petri dishes also containing LCA medium with 2 antibiotics (chloramphenicol 0.01% and streptomycin 0.01%). For each dilution, we made 5 replicates. All Petri dishes were incubated at 22 °C in dark conditions. We checked the plates daily for fungal growth, and we transferred individual mycelia to new LCA plates containing the same two antibiotics.

### 2.4. Quantification of Oxalotrophic Bacteria

To quantify the proportion of culturable oxalotrophic bacteria, we plated the microbial suspensions and the dilutions described above on a modified bi-layered AB Schlegel medium [35] with calcium oxalate (7 g·L^−1^), pH adjusted to 7, and with the fungicide cycloheximide 100 mg·L^−1^. For each dilution, 5 replicates were made, and all plates were incubated at 22 °C in the dark. The presence of halos around the colonies indicated oxalotrophic activity of the colony. We estimated the bacterial concentration by counting the number of colony-forming units (CFU) per gram of litter and using only one dilution of the suspensions (1/10,000).

### 2.5. DNA Extraction and Amplification of Taxonomic Marker

We extracted DNA from the mycelium of each fungal isolate using the ZR Fungal/Bacterial DNA MiniPrep kit (Zymo Research, Irvine, CA, USA) and following the manufacturer’s instructions. Subsequently, we targeted the 28S large subunit (LSU) rRNA gene as a fungal taxonomic marker. We used the primer pair LROR_F 5′-CCGCTGAACTTAAGCATATCAATA-3′ and LR5-F 5′-CGATCGATTTGCACGTCAGA-3′ to amplify a fragment of the LSU rRNA gene [36], with the following PCR scheme: one cycle of 95 °C for 4 min, then 35 cycles of 95 °C for 30 s, 56 °C for 30 s, and 72 °C for 90 s, ending with one cycle of 72 °C for 10 min. We performed PCR reactions in a total volume of 50 µL with the KAPA2G Robust kit (Kapa Biosystems, Wilmington, MA, USA). We purified the amplicons on Millipore MultiScreen microplates and then sent them for bidirectional Sanger sequencing at the GATC Biotech AG sequencing center (Germany). After quality trimming, we assembled forward and reverse sequences into contigs with SeqTrace version 0.9.0 [37]. Sequences have been deposited in GenBank under the accession numbers MW632957-MW633006.

### 2.6. Taxonomic Assignment and Phylogenetic Analysis

For each 28S rRNA gene sequence, we performed a BLASTN search [38] with both the nr/nt and the fungal 28S rRNA RefSeq curated databases (accessed on 2nd August 2020), and we collected top hits sequences. For each fungal class, we performed a phylogenetic analysis in order to refine the taxonomic assignment of each fungal strain, using the following procedure. We aligned sequences with SINA v1.2.11 [39] against the Silva LSU v132 database [40], and we filtered the resulting alignment using trimAl v1.4.rev22 with the gappyout method [41]. We used Smart Model Selection [42] to determine the best model of nucleic acid evolution of the filtered alignment based on the Akaike Information Criterion. Subsequently, we built a maximum-likelihood phylogenetic tree with PhyML 3.0 [43]. We calculated branch supports using a Chi2-based parametric approximate likelihood-ratio test (aLRT) [44]. Following a similar procedure, we also built a phylogenetic tree containing only the sequences from the 50 strains isolated in the present study, and we visualized it with Iroki [45].

### 2.7. Functional Characterization of the Fungal Strains

To describe the functional potential of the fifty isolated strains, we used ten selective media. First, we used the LCA medium (described above) as a minimal medium to measure fungal radial growth. We observed the fungal colonies daily and measured the diameter of the colonies until they reached the periphery of the plates. For each strain, the diameter of the colony (*n* = 3 replicates) was plotted over time, and a linear growth phase was identified. We estimated hyphal extension rate by linear regression of the linear growth phase [46]. We tested the ability of each fungus to grow on as well as to degrade cellulose and xylan using a CMC (carboxymethyl-cellulose) and a xylan media, respectively. CMC medium was composed of carboxymethyl-cellulose sodium salt (Sigma) 5 g·L^−1^ as sole source of carbon; K_2_HPO_4_ 1.0 g·L^−1^; (NH_4_)_2_SO_4_ 1.0 g·L^−1^; MgSO_4_∙7H_2_O 0.5 g·L^−1^; NaCl 0.5 g·L^−1^; agar 15 g·L^−1^; and pH was adjusted to 5 [47]. Xylan medium was composed of beechwood xylan (Sigma) 10 g·L^−1^ as the sole source of carbon; K_2_HPO_4_ 1.0 g·L^−1^; (NH_4_)_2_SO_4_ 1.0 g·L^−1^; MgSO_4_∙7H_2_O 0.5 g·L^−1^; NaCl 0.5 g·L^−1^; agar 15 g·L^−1^; and pH was adjusted to 5 [47]. The cellulolytic activity on the CMC medium and xylanolytic activity on the xylan medium were detected using 0.1% Congo red (Sigma) for staining for 40 min followed by washing with 1 M NaCl according to the Teather et al. method [48]. Ligninolytic activities were screened using the LCA medium containing 0.05% Remazol Brilliant Blue R (RBBR) (Sigma). Positive activities were indicated by the RBBR medium turning from blue to pale pink [49]. To investigate the in vitro production of calcium oxalate (Caox) crystals by the fungi, we used a malt (12 g·L^−1^) agar (15 g·L^−1^) medium supplemented with 5 g·L^−1^ CaCO_3_ [32]. A medium containing only agar (15 g·L^−1^) with 5 g·L^−1^ CaCO_3_ was also tested, but since not all the strains were able to grow on it, it was not included in the final analysis. We also investigated the dissolution of Caox by inoculating strains on a bi-layered AB Schlegel medium [35] supplemented with 3.2 g·L^−1^ Caox. Oxalate degradation by a fungal strain was indicated by the presence of a halo around the mycelium. We searched for the production of siderophores by the fungal strains by inoculating them on a chrome azurol S (CAS) agar medium prepared following the method of Tarnawski et al. [50]. The discoloration of the medium (blue to yellow or orange) indicated siderophore-producing fungal strains. We identified proteolytic activities by the presence of a halo around the mycelium of the strains inoculated on LCA medium supplemented with 5% commercial skimmed milk (950 g.kg^−1^, Migros) [51]. Finally, we investigated solubilization of inorganic phosphate by looking at the presence of a halo around the mycelium of the strains inoculated on NBRIP medium [52], composed of Ca_3_(PO_4_)_2_ 5 g·L^−1^; MgCl_2_∙6H_2_O 5 g·L^−1^; MgSO_4_∙7H_2_O 0.25 g·L^−1^; KCl 0.2 g·L^−1^; (NH_4_)_2_SO_4_ 0.1 g·L^−1^; glucose 10 g·L^−1^; agar 15 g·L^−1^; pH 7. All the media were autoclaved at 121 °C for 20 min.

For each strain, we inoculated *n* = 3 Petri dishes of each medium with a single fungal plug. Subsequently, we sealed them with Parafilm M and incubated them at 30 °C in dark conditions before analysis. In total, we analyzed 10 media × 50 strains × 3 replicates = 1500 Petri dishes.

### 2.8. Microscopic Observations

Using a malt (12 g·L^−1^) agar (15 g·L^−1^) medium supplemented with 5 g·L^−1^ CaCO_3_, we evaluated the production of Caox crystals for each fungal strain with a Leica DMR optical microscope and a ×400 magnification, following the procedure described in [32]. Because fungal strains grew at different rates, we observed the fungal colonies daily and sampled them for screening only when they just reached the periphery of the plates. For each plate, we observed both the youngest (at the periphery of the plate) and the oldest part (at the center of the plate) of the mycelia.

### 2.9. Data Analysis

We performed statistical analyses with R version 4.0.2 [53] and visualized data with the *ggplot2* package [54]. The existence of phylogenetic signal for the binary traits was tested using the *D* statistic [55], computed with the *caper* package (*phylo.d* function with 10⁵ permutations). A value of *D* = 0 indicates phylogenetically conserved traits as expected under a Brownian motion model, whereas *D* = 1 indicates randomly distributed binary traits. For the continuous trait, we used Pagel’s λ metric [56] with the *phytools* package (*phylosig* function) [57]. Here, λ = 0 indicates no phylogenetic signal, while λ = 1 indicates that traits evolved according to a Brownian motion model, and, therefore, traits show a strong phylogenetic signal.

## 3. Results

### 3.1. Evidence of an Active Oxalate-Carbonate Pathway (OCP)

Several conditions are required to establish the existence of an active OCP: the presence of both active oxalogenic and oxalotrophic organisms and a local increase in soil pH. All of which may eventually promote the precipitation of calcium carbonate (CaCO_3_). Concerning the first element, *T. indica* is known to be an oxalogenic plant in which calcium oxalate crystals have been observed in secondary xylem [58] and bark compartment [59] of the tree. The presence of culturable oxalotrophic bacteria in the litter was investigated by plate counting on artificial media. On average, we estimated a total of 5.96 10^7^ bacterial CFU.g^−1^ litter, including 1.76 10^7^ oxalotrophic CFU.g^−1^ litter (Figure 1A), which represents on average 29% of culturable oxalotrophic bacteria. The pH profiles revealed that pH was always higher under the tree (ranging from 7.5 to 8.4) than away from the tree (ranging from 6 to 7.2), indicating an alkalinization process under the influence of *T. indica* (Figure 1B). Finally, X-ray powder diffraction revealed the presence of quartz, calcite, and whewellite (calcium oxalate monohydrate) in the litter (Figure 1C). Weddellite (calcium oxalate dihydrate) was not detected. Overall, the results supported the existence of an active OCP in the study site.

### 3.2. Taxonomic Diversity

A total of fifty fungal strains were isolated from the litter of *T. indica*. Phylogenetic inference based on a fragment of the 28S rRNA gene of each strain (Appendix A) revealed a taxonomic diversity encompassing three phyla, namely Mucoromycota, Ascomycota, and Basidiomycota; and 23 genera (Figure 2). Fourteen genera were represented by a single strain and five genera by three strains. Additionally, four strains of *Pseudoseptoria*, five strains of *Absidia* and *Sirastachys*, as well as seven strains of *Aspergillus*, were isolated (Appendix A).

### 3.3. Functional Diversity 

Each fungal strain was further characterized by measuring eight binary and one continuous trait, using various selective media (Figure 2). All the strains were able to grow on all the selective media. Additionally, all the tests were performed in triplicate, and all results were always consistent among the three replicates. Regarding plant polymer degradation, CMC degradation was detected in 30 out of the 50 strains, while 36 strains were involved in xylan degradation. A total of 27 strains was involved in both CMC and xylan degradation. However, only one strain (strain 35), from the genus *Ceratobasidium* (Basidiomycota), was able to degrade RBBR that was used as a proxy for ligninolytic activity, and this strain also degraded xylan (Figure 2). Caox production was observed for nine strains (Figure 3), encompassing the three phyla. Caox degradation was detected for fifteen strains, and none of these strains were Caox crystal producers. Siderophore production and proteolysis were the two most widespread traits, found in 43 and 46 strains, respectively. Lastly, inorganic phosphate solubilization was observed for eight strains, all belonging to the Ascomycota. The continuous trait measuring the hyphal extension rate was highly variable, ranging from 9.4 (strain 46, *Bartalinia* genus) to 1709.7 µm h^−1^ (strain 09, *Neurospora* genus). No significant differences were observed between the distribution of hyphal extension rates within the different detected functions (Kruskall-Wallis, *p* = 0.377). For all these traits, we tested for the presence of a phylogenetic signal (Appendix A). A significant phylogenetic signal was detected for xylan degradation (*D* = 0.660, *p* = 0.044), Caox crystal production (*D* = 0.461, *p* = 0.016), Caox degradation (*D* = 0.429, *p* = 0.003), siderophore production (*D* = −0.058, *p* = 0.0001) and hyphal extension rate (λ = 0.999, *p* < 0.0001).

## 4. Discussion

We reported here the first evidence of an active OCP in Madagascar. To date, the OCP has been observed in numerous tropical and semiarid environments, with different tree species, and across various geographical areas, such as Israel [60], the USA [61], Bolivia [30], India [23], Cameroon [62], the Ivory Coast [13], Haiti, and Mexico [63]. Here we identified *T. indica* as an oxalogenic tree driving an active OCP. This species is relatively widespread in Madagascar [64], but more importantly, it is also present in Asia, Africa, and the Caribbean islands [65], suggesting that the OCP could also be detected there if soil properties are appropriate. Altogether, these results reinforce the idea that the OCP is a widespread phenomenon participating in the cycles of C and Ca in tropical and semiarid soils.

In the studied litter samples, the concentration of culturable oxalotrophic bacteria appeared to be two times higher than the one observed in OCP soil from Bolivia [30]. While culture-dependent approaches are known to be highly biased to estimate microbial diversity, the very high concentration of this functional group highlights its importance for soil ecology and biogeochemistry [14] and calls for further studies quantifying the diversity and abundance of oxalotrophic bacteria. Regarding the Caox crystals present in the litter, whewellite, but no weddellite was detected. A similar result was observed in soil samples from Bolivia [30] but, in African soils from active OCP, weddellite was detected [18]. Since fungi can produce both types of Caox crystals [32], this result is surprising, and further studies should include extensive microscopic observations and microanalyses of various samples to confirm this observation.

While the main goal of the present study was not to obtain a taxonomic overview of the fungal diversity associated with the OCP litter, we provide the first glimpse of this diversity via a culture-dependent approach. The described fifty strains represent 23 genera encompassing three phyla. With 42 strains, Ascomycota was the most abundant phylum in the dataset, which was also the case for studies investigating fungal litter diversity via metabarcoding approaches [66,67]. Among the Ascomycota, we also recovered strains belonging to genera that are globally widespread soil fungal generalists, such as *Fusarium*, *Trichoderma* and *Penicillium* [68]. Lastly, the only Basidiomycota isolated here, belonging to the *Ceratobasidium* genus, was also reported as an abundant member in decomposing leaf litter [67]. Altogether, the results suggest that the strains described here are representative members of the litter mycobiome, even if culture-independent surveys (i.e., metabarcoding and/or metagenomics) are still required to obtain a more comprehensive overview of the fungal diversity. Additionally, the use of more diverse culture media for isolation, with, for instance, different pH and different carbon and nitrogen sources, would allow recovering more diverse fungal strains. In combination with existing bacterial collections of oxalotrophic bacteria isolated from OCP [27], this fungal collection will offer a unique opportunity to design microcosm experiments [20] to better understand the mechanisms and regulations of the OCP. Indeed, fungal strains could be selected based on their functional potentials and hyphal extension rates described in this study.

To investigate the functional diversity of each fungal strain, we used various selective media, which are important tools for fungal ecology [69]. Because the strains were isolated from a decomposing litter, we first investigated the potential for plant polymer degradation, namely cellulose (CMC substrate), hemicellulose (xylan substrate), and lignin (RBBR substrate). More than half of the strains (*n* = 27) were identified as both CMC and xylan degraders, suggesting that these fungi can decompose the two most abundant plant carbohydrates (i.e., cellulose and hemicelluloses), potentially with synergistic metabolisms. For CMC degradation, the trait was not conserved at the genus level, with variation among strains of some genera, such as *Absidia*, *Pseudoseptoria*, *Aspergillus* and *Penicillium*. Regarding xylan degradation, a weak but significant phylogenetic signal was observed (*D* = 0.660, *p* = 0.044), indicating that this trait was not randomly distributed. The potential ligninolytic activity was observed for the only isolated Basidiomycota, *Ceratobasidium* sp. (strain 35). Interestingly, this genus has already been reported to be associated with leaf litter, and this presence was correlated with laccase and peroxidase activities [70], confirming its potential role in lignin degradation.

Proteolysis is known to be a widespread trait among soil microorganisms [71], and our data were no exception, with 92% of the strains presenting proteolytic activity. Such activity would allow the fungi to access an organic nitrogen source in an environment where C/N ratio can be high. Siderophore production was the second most abundant trait, being observed in 86% of the strains. It is also common among soil and litter fungi [72]. Indeed, in leaf litter, iron availability can be limited, and this element is essential for certain reactions, including the Fenton reaction involved in the fungal degradation of plant polymers [73]. Additionally, siderophores can be involved in mineral weathering in a more efficient way than low molecular mass organic acids, such as citrate or oxalate [74]. Saprotrophic fungi can contribute to phosphate solubilization [75] and thus improve soil fertility. Among our strains, only eight were able to solubilize inorganic phosphate, suggesting a minor contribution of the OCP fungi to the P cycle. This is in line with in situ measurements of phosphorus content in an OCP system, where this content was mostly influenced by soil depth but not by the OCP [62].

Regarding oxalate metabolism, we identified fifteen strains involved in Caox degradation and nine involved in Caox crystal production. Therefore, our results provide evidence that fungi associated with the OCP are not only involved in oxalate production but also in oxalate degradation, changing the current paradigm of the OCP in which oxalate degradation is mainly the result of bacterial activity. It has been shown in vitro that some white-rot fungi (Basidiomycota) can produce and dissolve calcium-oxalate crystals [32]. Here we demonstrated that other fungal taxa (Ascomycota and Mucoromycota) also possess these abilities. However, it is important to underline that no strain was able to both produce Caox crystals and degrade oxalate, suggesting functional complementarity among the fungal community involved in the OCP. For these two functions, we detected a significant phylogenetic signal, suggesting that these traits tend to be more present among close relatives than among distant relatives. Among the seven *Aspergillus* strains isolated here, four of them were able to produce Caox crystals, confirming previous reports about members of this genus [76]. Similarly, members of *Absidia* and *Penicillium* genera were identified as Caox producers, which was also previously reported [77,78]. However, to our knowledge, this is the first report of members of *Ceratobasidium* and *Ciliochorella* being able to produce Caox crystals. Regarding the fungi able to degrade oxalate, we found one strain of *Fusarium* and *Trichoderma*, two genera for which this capacity has already been reported [79]. Interestingly, we also reported for the first time in vitro the ability to decompose calcium oxalate for strains belonging to the genera *Absidia*, *Bartalinia*, *Lichtheimia*, *Nigrosabulum*, *Ochroconis*, *Pseudocoleophoma*, *Purpureocillium* and *Sirastachys*, and thus, we extended the knowledge on the diversity of this functional group [31]. Further studies could focus on the regulation and the quantification of these metabolisms.

## 5. Conclusions

This study provides the first insights into the taxonomic and functional diversity of the litter fungi associated with an active OCP in Madagascar under a *Tamarindus indica* tree. In addition to the capacity to decompose cellulose and hemicellulose, the characterization of fifty strains revealed the importance of fungi in the OCP for siderophore production, proteolysis, Caox crystal production but also surprisingly for oxalate degradation. These results provide new elements to better understand the role of fungi in the OCP. 

## Figures and Tables

**Figure 1 microorganisms-09-00985-f001:**
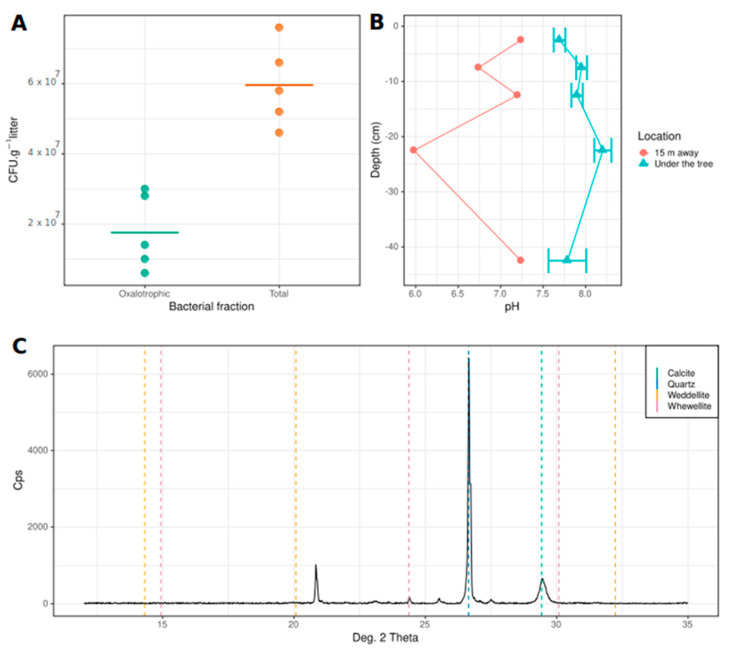
Confirmation of the existence of an active oxalate-carbonate pathway (OCP) associated with the tree *Tamarindus indica*. (**A**). Concentrations of total and oxalotrophic culturable bacteria from *Tamarindus indica* litter. Horizontal lines represent the mean of the distributions. (**B**). pH profiles under and 15 m away from the tree. Bars represent the standard error of the mean (*n* = 3). (**C**). X-ray diffractogram of the *T. indica* litter revealing the presence of quartz, calcite, and whewellite.

**Figure 2 microorganisms-09-00985-f002:**
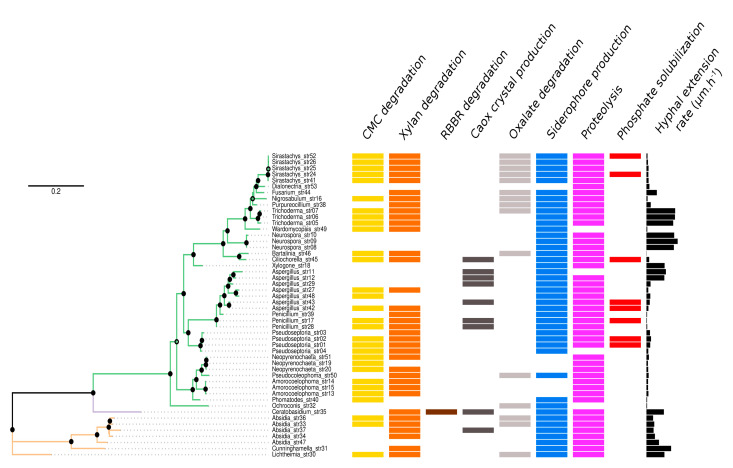
Functional traits of the fungal strains as a function of their phylogenetic relationships. Branches of the maximum-likelihood tree are colored by phylum (Ascomycota in green, Basidiomycota in mauve, and Mucoromycota in brown). Filled dots represent Chi2-based parametric approximate likelihood-ratio test values ≥ 0.9 and open dots values ≥ 0.7. The phylogenetic tree includes 50 strains encompassing 23 genera: *Aspergillus* (*n* = 7), *Absidia* (*n* = 5), *Sirastachys* (*n* = 5), *Pseudoseptoria* (*n* = 4), *Amorocoelophoma* (*n* = 3), *Neopyrenochaeta* (*n* = 3), *Neurospora* (*n* = 3), *Penicillium* (*n* = 3), *Trichoderma* (*n* = 3), *Bartalinia* (*n* = 1), *Ceratobasidium* (*n* = 1), *Ciliochorella* (*n* = 1), *Cunninghamella* (*n* = 1), *Dialonectria* (*n* = 1), *Fusarium* (*n* = 1), *Lichtheimia* (*n* = 1), *Nigrosabulum* (*n* = 1), *Ochroconis* (*n* = 1), *Phomatodes* (*n* = 1), *Pseudocoleophoma* (*n* = 1), *Purpureocillium* (*n* = 1), *Wardomycopsis* (*n* = 1) and *Xylogone* (*n* = 1).

**Figure 3 microorganisms-09-00985-f003:**
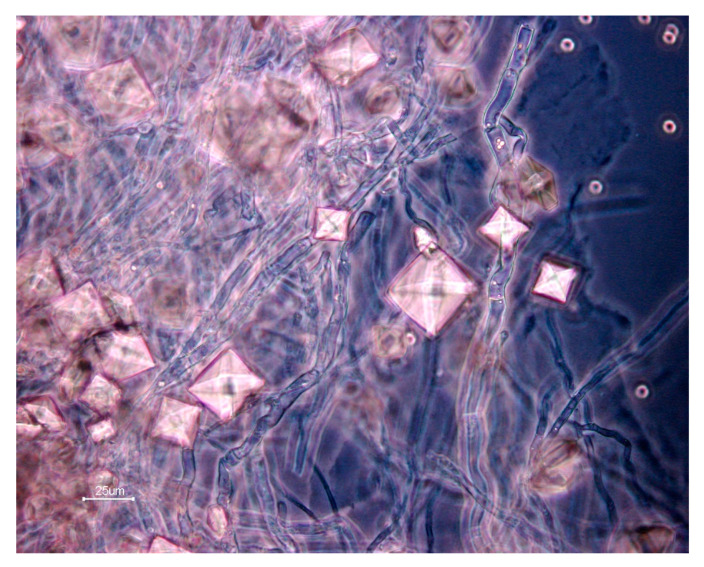
Example of phase-contrast microscopy image of calcium oxalate crystals produced by *Aspergillus* sp. (strain 11).

## Data Availability

Sequences of the 28S rRNA genes have been deposited in GenBank under the accession numbers MW632957-MW633006.

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
