# Peer review of "Functional Diversity of the Litter-Associated Fungi from an Oxalate-Carbonate Pathway Ecosystem in Madagascar"

_microorganisms, 2021, doi:10.3390/microorganisms9050985_

Round 1

Reviewer 1 Report

In this manuscript, the authors describe a study where they isolated and characterized fungi and their potential roles in the oxalate-carbonate pathway. Overall I enjoyed reading the manuscript and thing that it is largely well-done and useful in expanding our understanding of the roles of fungi in biogeochemical cycles.

My main criticism is that the authors overstate the impact of this work, just a little bit, especially with the first paragraph of the discussion, and in trying to say anything about the abundance of the bacteria or fungi involved in this process. There was no attempt to overcome biases in isolation, and in fact, the authors did their initial isolation of fungus and bacteria with highly selective media. As such, we really cannot conclude anything about the proportion of the microbial community that is involved with OCP here. I don’t think this is something that should warrant rejection of the manuscript, but I do think that the authors need to pull back from discussions of abundance or diversity.

Other than the comments below, I think that the discussion was particularly well done.

Please see below for more specific comments in relation to sections of the manuscript.

line 33-35 mentions that the results challenge the current view of the OCP, but if that’s the case, it would be better if that current view was mentioned at the beginning of the abstract. As it stands, this is too vague because I don’t know what view is being challenged.

line 75 - sampling was done in Madagascar, yet there is no mention of permits. Were permits not required? Either way, a mention of the permits or that they were not required, here or in the acknowledgements would be good.

lines 75-81 - I would like more detail about the litter sampling. How much material was taken? What steps were taken to prevent contamination of the litter from soil or the researchers?

Methods, in general, would be clearer and easier to read if they were written in a more active voice. This is a personal preference, so I don’t think it’s super vital, but I do strongly believe that it makes writing better.

line 94 - replace ‘during’ with ‘for’

line 95 - define LCA

line 104 - did the new LCA plates also contain antibiotics, or were these only used for the initial isolation?

lines 92-104 - why was a pH of 5 used for the media? In figure 1B, we can see that the natural pH of the soil was between 6 and 8, leaning toward the basic under the tree, yet the media used to grow the fungi was acidic. I wonder how many fungi that naturally occur in the litter would not grow at this pH. Do you have any measure of what proportion of fungi that actually occur in the environment were isolated? What happens if you provide a variety of media at different pH’s? Also, litter pH wasn’t measured - why?

line 107-113 - what pH were the bacteria grown at?

Figure 1 - why are there no error bars for the soil samples from 15 m away?

line 229-254 - I feel that it is odd and not very informative to put the strain counts in a plot like this. I would much prefer a phylogeny, that demonstrates the diversity and numbers of strains instead.

line 265-266 - affiliated is not the right word here. Rephrase, as: “Only strain 35, from the genus Ceratobasidium (Basidiomycota), was able…

line 298 - I really like Figure 3. I wish that the authors would expand this figure so that the phylogeny could be more informative. The information in Figure 2 could be captured in the phylogeny. As well, I think it would be better to include some sequences from sister clades to put the phylogeny in context.

line 329-337 - It is puzzling to me that the authors would start their discussion with this paragraph. Is there any reason to expect that OCP would not be ubiquitous? I just don’t think that this is carrying the novelty that the authors are suggesting, in terms of the geographic distribution of this process. If I am wrong on this point, please expand and explain here why we would not expect this.

line 339 - were the same isolation methods used in the Bolivia study? Is this comparison valid? You were clearly selecting for oxalotrophic bacteria with this study. In order to say anything about abundance, you would need to compare the proportion of oxalotrophic bacteria to the total diversity of bacteria found using general rich media.

Reviewer 2 Report

The manuscript "Characterization of litter-associated fungi revealed their potential role as both oxalate producers and degraders in the oxalate-carbonate pathway" by Hervé et al. contains interesting data, but makes a chaotic and strange impression. Throughout the text, the authors focus on the oxalate-carbonate pathway, but the results have little to do with the study of this process.

The introduction is entirely devoted to describing OCP. But the investigation includes other functional characterizations of the fungi. The relevance of these studies is not substantiated.

Research methods are not detailed enough. Cultivation on selective media and microscopy should be described separately. The time of cultivation as well as the conditions for sterilization of the media are not indicated.

Have the morphology of fungal cultures been investigated?

(Line 167) "Oxalotrophic activity of the fungal strain ...." As far as I know, there is no evidence of oxalotrophic activity of fungi. Fungi have enzymes (oxalate oxidase and oxalate decarboxylase) that degrade oxalate, but in fungi have not described ways of assimilating oxalic acid as a source of carbon and energy as in oxalotrophic bacteria. Whether fungi can exhibit oxalotrophic activity is a controversial question, and there are no detailed experimental studies on this topic in this manuscript. The ability of fungi to break down calcium oxalates is not evidence of their oxalotrophic activity. The same refers to the "Discussion" section.

In general, the complex of methods used for the physiological and biochemical study of fungi is very superficial. The authors did not use any analytical method.

Section "Results", "3.1 Evidence of an active oxalate-carbonate pathway (OCP)". The data given in this section does not correspond to its title. The detection of quartz, calcite, calcium oxalate and organisms potentially producing and consuming oxalic acid is not at all evidence of the OCP biochemical processes in the system. Have oxalotrophic bacteria been identified? Is the data about bacteria presented anywhere?

The functional diversity data is poorly illustrated. Table S1 contains only qualitative data, but no quantitative indicators.

The quality of the picture of the calcium oxalate crystals is rather poor. The granes of the weddellite bipyramid is not clearly visible. How was oxalate determined in cultures of fungi? Only by morphology? Did weddellite and whewellite differentiate? On what day of fungal growth were the results of the experiment are presented? The ratio of weddellite to whewellite can vary significantly over time, so it is important to know the exposure time.

 The manuscript describes well the biodiversity of fungi. But for some reason the authors themselves write that the study of biodiversity was not included in the main tasks of the study. At the same time, the main aim of the study has not been formulated at all. Since the significance of the results in physiology and biochemistry is very small, in my opinion, the manuscript may be important precisely in the aspect of assessing the biodiversity of fungi of this biocenosis with brief ecological and physiological characteristics of fungi. But it should be a completely different article with a different title, different accents, and a different discussion.

Round 2

Reviewer 2 Report

The authors revised the manuscript in detail and made it acceptable for publication.
As the only comment, I would like to ask the authors to  breafly formulate the main of the study in 1-2 sentences in the end of introduction order to make the manuscript clearer and more understandable.

Author Response

The authors revised the manuscript in detail and made it acceptable for publication.

As the only comment, I would like to ask the authors to briefly formulate the main goal of the study in 1-2 sentences in the end of introduction in order to make the manuscript clearer and more understandable.

Reply - As requested, we have now added at the end of the introduction, a sentence describing the main goals of the study.

Lines 72-75 : “Our study aims at: (i) isolating and phylogenetically identifying fungi associated with an active OCP, (ii) characterizing the functional potential of these fungal strains, especially regarding their contributions to C, N, P, Ca and Fe cycles, and (iii) clarifying the role of fungi in the functioning of the OCP.”